# Unmodulated 40 Hz Stimulation as a Therapeutic Strategy for Aging: Improvements in Metabolism, Frailty, and Cognitive Function in Senescence-Accelerated Prone 10 Mice

**DOI:** 10.3390/biom14091079

**Published:** 2024-08-28

**Authors:** Tatsunori Shimizu, Hidetaka Ota, Ayuto Kodama, Yasuhiro Suzuki, Takako Ohnuma, Rieko Suzuki, Kaoru Sugawara, Yasushi Sato, Hiroyuki Kodama

**Affiliations:** 1Advanced Research Center for Geriatric and Gerontology, Akita University, Akita 010-8543, Japan; tatsunori@med.akita-u.ac.jp (T.S.); ay-kodama@med.akita-u.ac.jp (A.K.); ohnuma@med.akita-u.ac.jp (T.O.); kaoruko02@jimu.akita-u.ac.jp (K.S.); 2Department of Occupational Therapy, Graduate School of Medicine, Akita University, Akita 010-8543, Japan; 3Department of Complex Systems Science, Graduate School of Information Science, Nagoya University, Nagoya 464-8601, Japan; ysuzuki@i.nagoya-u.ac.jp; 4Facetherapie Co., Ltd., Tokyo 150-0002, Japan; 5555rieko@gmail.com; 5Take Out Air Inc., Tokyo 106-0032, Japan; info@takeoutair.jp; 6Social Medical Corporation of Seiwakai, Akita 018-1401, Japan; kodamah@seiwakai-net.com

**Keywords:** aging, glucose tolerance, muscle, unmodulated 40 Hz stimulation

## Abstract

With aging populations in many countries, including Japan, efforts to mitigate the aging-related decline in physical function have gained importance not only for improving individual quality of life but also for mitigating the effects of this loss of function on society. Impaired glucose tolerance, muscle weakness, and cognitive decline are well-known effects of aging. These interrelated factors can create a vicious cycle because impaired glucose tolerance can accelerate muscle weakness and cognitive decline. Unmodulated 40 Hz (u40Hz) stimulation is imperceptible to the human ear and has been reported to improve cognitive function in humans and mice. However, research on the effects of u40Hz stimulation is still limited. This study aimed to report the effects of u40Hz stimulation on glucose tolerance and muscle strength in senescence-accelerated prone (SAMP)-10 mice, a model of accelerated aging. SAMP-10 mice underwent five weeks of u40Hz stimulation followed by glucose-tolerance tests, cognitive and behavioral assessments, and frailty evaluations. In comparison with the control group, the u40Hz-stimulation group showed mitigation of age-related decline in glucose tolerance, a better frailty index (FI), and notably preserved muscle strength. Microarray analysis of stimulated muscle tissue revealed significant upregulation of β-oxidation genes and genes functioning downstream of peroxisome proliferator-activated receptor gamma, and significant downregulation of clock genes. These findings indicate the beneficial effects of u40Hz stimulation on glucose tolerance, muscle strength, and cognitive function, warranting further research in this area.

## 1. Introduction

With the aging of the population in numerous countries, extending healthy life expectancy has become a crucial issue. With people living longer, maintaining a high quality of life has become as important as increasing lifespan. Frailty is defined a dynamic state wherein individuals experience loss in one or more domains of human functioning (physical, psychological, or social). Frailty is influenced by a number of variables, and the occurrence of frailty increases the risk of adverse outcomes. However, assessments of the relationship between aging and frailty should move beyond viewing aging as a natural process and attempt to understand its broader impact. Age-related declines in physical function and metabolic capacity may increase the risk of frailty. Moreover, in addition to decreasing the individual’s quality of life, this decline also results in an increase in the societal economic burden due to the rising need for caregiving and support.

Muscle weakness is an important factor in the diagnosis of frailty [1], and muscle strength is a key indicator of frailty in clinical practice. Muscle strength is an essential element in maintaining the ability to perform daily activities and a favorable quality of life. Preservation of muscle strength is extremely important for maintaining a healthy lifestyle. Muscle strength tends to decline with age, and the incidence of sarcopenia (progressive decline in muscle mass, strength, and physical function) is increasing in Asia, where the elderly population is increasing [2]. Sarcopenia has recently received increased attention in clinical settings owing to its association with various adverse health effects such as decreased mobility and increased mortality rates [3]. Currently, the most effective strategy for preventing sarcopenia is to adopt a healthier lifestyle, including adherence to a high-quality diet and regular physical activity [4]. However, this may be difficult in certain living environments, and identification of additional preventive measures is important in this regard.

Diabetes mellitus is a significant global health challenge with an increasing prevalence worldwide. Complications associated with diabetes, including cardiovascular disease, neuropathy, nephropathy, and retinopathy, pose substantial risks to the affected individuals and contribute to increased morbidity and mortality. The decline in glucose tolerance with age is well known [5,6], and the prevalence of diabetes has been shown to increase with age [7]. The causes of the age-related impairment in glucose tolerance include decreased insulin sensitivity and islet cell dysfunction [8]. Possible reasons for the diminished effectiveness of insulin with aging include (1) increased abdominal fat mass, (2) decreased physical activity, (3) sarcopenia, (4) mitochondrial dysfunction, (5) hormonal changes (such as reduced levels of insulin-like growth factor 1 and dehydroepiandrosterone), and (6) heightened oxidative stress and inflammation [9]. Moreover, a decrease in muscle strength and mass can cause impaired glucose tolerance, and elderly diabetic patients have been reported to show decreased muscle strength and mass [10]. Furthermore, diabetes is a risk factor for dementia [11].

Unmodulated 40 Hz (u40Hz) stimulation is a type of tactile stimulation that utilizes ultra-low-frequency vibrations, typically at 20 Hz or lower, that fall within the non-audible range for humans [12]. These vibrations are believed to engage various levels of the central nervous system, and this stimulation elicits cognitive responses in humans [13]. In a study by Clements-Cortes et al., both 40 Hz acoustic stimulation and visual stimulation with Digital Versatile Discs were applied as interventions for patients with mild AD and outpatients in a healthcare facility [14]. However, none of the previous reports have described the effects of u40Hz stimulation on muscle strength or glucose tolerance. Vibration stimulation has been reported to promote the healing of muscle injury [15] and alleviate acute exercise fatigue [16], suggesting that vibration stimulation may have restorative and protective effects on muscles.

Although u40Hz stimulation is thought to affect the nervous system primarily through the effects of tactile stimulation on the central nervous system, the fact that u40Hz stimulation can transmit vibrations deep into the body suggests that its effects may be directly involved in muscle contraction and strengthening. Therefore, we hypothesized that u40Hz stimulation has both direct effects on muscles and indirect effects on glucose tolerance via muscles, and we aimed to test this hypothesis using senescence-accelerated prone (SAMP)-10 mice, a model of accelerated aging.

## 2. Materials and Methods

### 2.1. Animals

Male SAMP-10 mice (age, 15 weeks) were purchased from Japan SLC, Inc. (Shizuoka, Japan) and housed and maintained in a room at 22 ± 2 °C with automatic light cycles (12 h light/dark) and relative humidity of 40–60%. The mice had free access to a normal diet (CE-2; Clea Japan, Tokyo, Japan) and water. Animal experiments were conducted in accordance with the National Institutes of Health Guide for the Care and Use of Laboratory Animals (8th edition, 2011) and approved by the Research Ethics Committee of the Faculty of Medicine, Akita University. The mice were divided into one group (*n* = 11) that underwent 5 weeks of u40Hz stimulation and a control group (*n* = 12) that did not undergo u40Hz stimulation. Before and after 5 weeks of u40Hz stimulation in the u40Hz-stimulation group, mice in both groups underwent the glucose-tolerance test; the Y-maze and open-field tests for cognitive and behavioral assessments; and frailty index (FI) evaluations for assessment of frailty as a measure of muscle strength, and the results of the two groups were compared.

### 2.2. U40Hz Stimulation

Mice in the u40Hz-stimulation group received continuous vibration stimulation and inaudible low-frequency stimulation at 40 Hz for 24 h a day for 5 weeks. The u40Hz stimulation was administered using a u40Hz-stimulation system, which consisted of an MP3 player (RUIZU^®^ Digital Player X02 (Shenzhen RUIZU Technology Co., Ltd., Shenzhen, China)) and woofer (YAMAHA-NS-SW050/B (Yamaha Corporation, Hamamatsu, Shizuoka, Japan)).

### 2.3. Oral Glucose-Tolerance Test

Glucose and plasma insulin levels were measured before and after the oral administration of glucose (2 g/kg body weight) following a 16 h fast. Blood glucose levels were measured at pre-glucose load and 15, 30, 60, and 120 min post-load using the Glutest Mint assay (Sanwa Kagaku Kenkyusho, Aichi, Japan). Plasma insulin concentrations were measured at pre-glucose load and 30 min post-load using an ultrasensitive mouse insulin enzyme-linked immunosorbent assay kit (Morinaga Institute of Biological Science, Kanagawa, Japan).

### 2.4. Food Intake Experiment

To assess food intake, we measured the reduction in the amount of food over a three-day period. The experimental procedure was as follows: Initially, we measured and recorded the weight of the food provided to each mouse at the beginning of the experiment. The mice were then allowed to consume food freely for a period of three days. At the end of this period, we weighed the remaining food. The total amount of food consumed over the three days was calculated by subtracting the final weight from the initial weight. This total was then divided by three to obtain the average daily food intake for each mouse.

### 2.5. FI Assessment

Each mouse was assigned an FI score based on the 31-item FI scale [17]. The severity of each item was rated on a scale of 0, 0.5, or 1. The FI score was determined by averaging the ratings assigned by three evaluators for each item. Finally, the total score of all 31 items was divided by 31 to calculate the FI.

### 2.6. Physical Function Measurements

The forelimb grip strength of the SAMP-10 mice was measured using a Grip Strength Meter (Columbus Instruments, Columbus, OH, USA). Locomotor function, including travel distance and average walking speed, were evaluated in SAMP-10 mice of the same age using the open-field test.

### 2.7. Open-Field Test

To evaluate locomotion and signs of depression, a modified open-field test was used to measure the fear response to novel stimuli. This test was performed using an open-field system (O’Hara Co., Ltd., Tokyo, Japan) with a 10 cm area adjacent to the surrounding wall designated as the periphery and the remainder of the open field constituting the central area. The dimensions of the open field were 500W × 500D × 300H mm^3^. The SAMP-10 mice were allowed to freely explore the open-field environment for 5 min. Parameters such as distance traveled, the ratio of distance traveled in the central area to the total distance traveled, time spent in the center, and alternation rate were analyzed. The software used for the open-field test was TimeOFCR1 (O’Hara & Co., Ltd.).

### 2.8. Y-Maze Test

The Y-maze test (O’Hara Co., Ltd., Tokyo, Japan), a task assessing hippocampal-dependent short-term spatial working memory and reference memory, was conducted to evaluate the exploratory behavior of the SAMP-10 mice. This test involved observing the behavior of the mice, including the number and sequence of entries into each arm, for 5 min. An entry was recorded when the mouse’s entire body was inside the arm. The mice were considered to show alternation behavior when they entered the three different arms consecutively, reflecting their working memory capacity. The number of spontaneous alternation behaviors was counted, and the alternation ratio was calculated as the number of spontaneous alternations/(the total number of arm entries − 2). The dimensions of the Y-maze were as follows: corridor floor width = 30W mm, corridor upper part width = 120W × 400D × 120H mm The software used for the Y-maze test was TimeYM1 (O’Hara & Co., Ltd.).

### 2.9. Microarray Analysis

The triceps brachii muscle from SAMP-10 mice was used for microarray analysis. Briefly, total mouse RNA was extracted using the RNeasy Mini Kit (Qiagen, Hilden, Germany). Target RNA was prepared by converting the mRNA into double-stranded cDNA using a T7-(dT)24 primer incorporating a T7 RNA polymerase promoter. Double-stranded cDNA synthesis and generation of biotin-labeled cRNA were performed with the GeneChip^®^ WT Plus Kit (Affymetrix, Thermo Fisher Scientific, Santa Clara, CA, USA). After fragmentation of cRNA to sizes ranging from 35 to 200 bases by heating (35 min at 95 °C), 10 μg of RNA fragments was hybridized (16 h at 45 °C) to the Affymetrix Mouse Clariom S Array (Affymetrix, Thermo Fisher Scientific). After hybridization, the gene chips were automatically washed and stained with streptavidin–phycoerythrin using a GeneCip Fluidis Station 450. The chips were scanned using a GeneChip Scanner 3000 7G (Affymetrix, Thermo Fisher Scientific, Inc.). Primary data analysis was performed using the Affymetrix Transcriptome Analysis Console (TAC) software, version 4.0. 3.14 and normalized and analyzed in accordance with the TAC user guide. Data were deposited in the National Center for Biotechnology Information Gene Expression Omnibus (accession number: GSE269209).

### 2.10. Statistical Analyses

Statistical analyses were performed using a two-way repeated measures ANOVA to evaluate the effects of u40Hz stimulation and time on blood glucose levels. Welch’s *t*-test was used for comparison of data from the oral glucose-tolerance test, whereas the Mann–Whitney U test was used to analyze the other results. The results were presented as mean ± standard deviation for the oral glucose-tolerance test and median ± interquartile range for the other analyses. Statistical significance was set at *p* < 0.05. All analyses were performed using SPSS (Version 26.0, SPSS Inc., Chicago, IL, USA).

## 3. Results

### 3.1. U40Hz Stimulation Improves Glucose Tolerance in Aging SAMP-10 Mice

The glucose tolerance of SAMP-10 mice, a mouse model of accelerated aging, was tested at 25 weeks of age and then at 31 weeks of age after 5 weeks of u40Hz stimulation. In the oral glucose-tolerance test after u40Hz stimulation, the u40Hz-stimulation group (*n* = 11) showed significantly lower blood glucose at fasting (65.5 ± 18.9 mg/dL vs. 101.9 ± 26.2 mg/dL), 15 min post-load (236.1 ± 62.5 mg/dL vs. 346.8 ± 75.6 mg/dL), and 30 min post-load (285.8 ± 78.5 mg/dL vs. 368.8 ± 77.3 mg/dL) than the control group (*n* = 12) (Figure 1). Before the u40Hz stimulation, the two-way ANOVA indicated only a significant interaction effect with time between the two groups (*p* < 0.001). After the u40Hz stimulation, we observed a significant main effect due to the u40Hz stimulation (*p* = 0.028) and a significant interaction effect with time (*p* < 0.001). The combined interaction effect of u40Hz stimulation and time approached significance (*p* = 0.053). Body weight, food intake, and body temperature did not differ significantly between the two groups before and after stimulation. Similarly, the insulin levels remained comparable between the groups before and after glucose loading.

### 3.2. U40Hz Stimulation Prevents Age-Related Muscle Weakness and Significantly Improves FI

Measurements of FI in SAMP-10 mice were obtained at 25 weeks and then at 31 weeks after 5 weeks of u40Hz stimulation. While the two groups showed no significant difference in FI before stimulation, the u40Hz-stimulation group (*n* = 11) showed a significantly lower FI than the control group (*n* = 12) after u40Hz stimulation (Table 1, Figure 2). Among the parameters used to evaluate the FI (2.25 ± 1.00 vs. 3.23 ± 1.43), grip strength (144.1 ± 28.3 gf vs. 91.7 ± 34.6 gf), coat condition, and piloerection were better in the u40Hz-stimulation group than in the control group.

### 3.3. U40Hz Stimulation Could Potentially Improve Spatial Memory and Learning Skills

We conducted open-field and Y-maze tests to assess mouse activity, stress, cognitive function, and spatial memory. In the open-field test, the two groups showed no significant differences in the total distance, average speed, or time spent in the center region (Figure 3), indicating equivalent activity and stress levels. However, in the Y-maze test, while the two groups showed no significant difference in alternation behaviors before u40Hz stimulation, the u40Hz-stimulation group (*n* = 11) showed significantly better results than the control group (*n* = 12) after stimulation (66.7% ± 50.0% vs. 25.0% ± 44.1%; Figure 3), indicating the potential beneficial effect of u40Hz stimulation on spatial memory and learning skills.

### 3.4. U40Hz Stimulation Results in Upregulation of Genes Associated with the PPARγ Pathway and β-Oxidation Pathway While Downregulating the Expression of a Clock Gene

To investigate the mechanisms underlying the effects of u40Hz stimulation on glucose tolerance and muscle strength in SAMP-10 mice, microarray analysis using biceps brachii muscle specimens was conducted to compare the differences in gene expression between groups with and without u40Hz stimulation (Appendix A). The u40Hz-stimulation group showed increased expression of genes related to fatty acid uptake downstream of peroxisome proliferator-activated receptor gamma (PPARγ) signaling (*Lpl*, *Slc27a1*, *Acsl1*) and genes involved in fatty acid oxidation (*Acadl*) (Appendix A). Furthermore, the u40Hz-stimulation group showed increased expression of genes involved in β-oxidation within the mitochondria (*Slc25a20*, *Acadl*, *Hadhb*) (Appendix A), suggesting that u40Hz stimulation may promote energy production in the mitochondria. However, the expression of *Arntl*, a major clock gene, was significantly decreased (Appendix A).

## 4. Discussion

This study demonstrated that u40Hz stimulation can mitigate some of the deleterious effects of aging in SAMP-10 mice, a model of accelerated aging. Our findings revealed multifaceted benefits of u40Hz stimulation, including improved glucose tolerance, reduced frailty, enhanced cognitive function, and significant changes in gene expression related to metabolism.

The results of glucose-tolerance tests indicated that u40Hz stimulation significantly improved glucose handling in aged SAMP-10 mice. Specifically, the u40Hz-stimulation group exhibited lower blood glucose levels than the control group during fasting and in the glucose-tolerance test. This effect was observed without significant differences in body weight, food intake, or insulin secretion, suggesting that the primary mechanism may involve enhanced glucose utilization and insulin sensitivity in the peripheral tissues. The upregulation of genes associated with the PPARγ pathway and β-oxidation supports this hypothesis. These pathways are critical for lipid metabolism and energy production and can enhance the efficiency of glucose metabolism and contribute to the observed improvements in glucose tolerance. PPARγ is a nuclear receptor involved in the control of metabolism that is predominantly found in adipose tissue and plays a significant role in adipocyte differentiation, which has led to the belief that its effects in adipose tissue are vital for explaining its role in insulin sensitization. However, recent studies have also emphasized the importance of its function in other tissues [18]. The expression of *Pparg* in muscles has been reported to be induced by an increase in reactive oxygen species due to exercise [19]. Additionally, the activation of AMP-activated protein kinase by exercise has been reported to increase the expression of PPARγ1, an isoform of PPARγ [20]. Sasaki et al. also demonstrated that PPARγ1 enhances the expression of lipoprotein lipase, which is considered an adaptation that promotes the uptake of fatty acids into muscle during exercise. Similarly, u40Hz-stimulated mice in this study showed an increase in *Lpl* expression, suggesting that u40Hz stimulation may act as an exercise-like stimulus on the muscle. Other genes downstream of PPARγ, such as *Slc27a1*, *Acsl1*, and *Acadl*, were also upregulated. Using mice with muscle-specific overexpression of *Slc27a1*, Holloway et al. [21] reported that *Slc27a1* increases the rate of transport of long-chain fatty acids into the muscle and redirects these lipids to oxidation rather than accumulation in the muscle. However, when examined in a high-fat diet, the insulin resistance of these *Slc27a1*-overexpressing mice did not differ from that of wild-type mice. Zhao et al. [22] reported that *Acsl1* deficiency resulted in an overall defect in muscle fuel metabolism and increased protein catabolism, causing exercise intolerance, muscle weakness, and muscle cell apoptosis. Long-chain acyl-CoA dehydrogenase (ACADL) is a key enzyme in mitochondrial fatty acid oxidation [23], catalyzing the initial step of β-oxidation of long-chain fatty acyl-CoAs. Mice with ACADL deficiency exhibit hepatic insulin resistance caused by impaired fatty acid oxidation [24]. In addition, since the expression of many genes in the β-oxidation pathway is elevated, the movement of these genes, taken as a whole, promotes fatty acid uptake into muscle and fatty acid oxidation in the u40Hz-stimulation group. This is similar to the response to exercise seen in healthy muscle [25] and contrasts with the markedly higher rate of fatty acid uptake and the corresponding increase in intracellular lipid accumulation in human muscle with insulin resistance, which is not accompanied by fatty acid oxidation [26]. Targeting fatty acid uptake and oxidation has been suggested to be a potential therapeutic approach for the treatment of insulin resistance [24], which may explain the improvement in glucose tolerance observed in this study.

The FI assessments in our study indicated that u40Hz stimulation significantly attenuated the progression of frailty in SAMP-10 mice. While the two groups of mice showed no differences before the u40Hz stimulation, after 5 weeks of stimulation, the u40Hz-stimulation group exhibited better grip strength, coat condition, and reduced piloerection. These findings indicate that u40Hz stimulation may help preserve muscle function and overall physiological condition in aging mice. A decline in the fatty acid oxidation capacity of muscles has been observed with aging [27]. Defects in fatty acid oxidation can cause chronic progressive muscle weakness [28]. The increased expression of genes involved in fatty acid oxidation within the mitochondria observed in the microarray analysis suggests that u40Hz stimulation enhances mitochondrial function and energy production, potentially facilitating the maintenance of muscle strength and reducing frailty.

u40Hz stimulation also showed clear cognitive benefits in the Y-maze test results, wherein u40Hz-stimulated mice showed significantly better spatial memory and learning performance than the controls. Although the open-field test indicated equivalent activity and stress levels between the groups, the improvement in the Y-maze test suggests that u40Hz stimulation may have specific effects on cognitive function. This result confirms previous reports showing improvements in cognitive performance with u40Hz stimulation [14].

Gene expression analysis provides insights into the molecular mechanisms underlying the beneficial effects of u40Hz stimulation. The upregulation of genes involved in the PPARγ and β-oxidation pathways indicates enhanced lipid metabolism and mitochondrial function. This could improve overall energy efficiency and reduce the metabolic burden associated with aging. The downregulation of *Arntl*, also known as Brain and Muscle ARNT-Like Protein-1 (*Bmal1*), a clock gene, may indicate an interaction between u40Hz stimulation and circadian regulation, which could influence various physiological processes, including metabolism and cognitive function. Previous studies using deletion [29] and overexpression [30] of *Bmal1* in beta cells have shown that *Bmal1* plays an important role in glucose tolerance and insulin secretion. On the other hand, obesity has also been reported to be associated with higher expression levels of *Bmal1* in peripheral blood mononuclear cells [31], suggesting that the downregulation of *Bmal1* expression due to u40Hz stimulation may result from metabolic improvements. Additionally, the function of BMAL1 varies across different tissues; for instance, mice with intestine-specific BMAL1 knockout show resistance to obesity when fed a high-fat diet [32] Further research is required to clarify this point.

The novelty of our findings lies in the application of u40Hz stimulation beyond its well-documented effects on cognitive function improvement and amyloid-beta clearance through aquaporin-4, as recently reported [33]. Our study is the first to demonstrate the positive impact of u40Hz stimulation on glucose tolerance and muscle preservation, which are significant aspects of aging that have not been previously explored. However, there are several limitations to our study. We used SAMP-10 mice, which are a model of accelerated aging. It will be essential to verify these findings in naturally aging mice, frail mice, and diabetic model mice in future research. Additionally, while u40Hz stimulation has shown benefits in other studies, the effects of other lower frequencies should be investigated. Furthermore, the results of our microarray analysis need to be validated through detailed cellular-level studies.

## 5. Conclusions

Stimulation with u40Hz appears to be a promising intervention since it mitigated some aspects of aging in SAMP-10 mice. The improvements in glucose tolerance, frailty, and cognitive function, coupled with the significant changes in the expression of genes related to metabolism, suggest that u40Hz stimulation could enhance overall health and longevity. However, further studies are needed to elucidate the precise mechanisms underlying the effects of u40Hz stimulation and explore the potential for translating these findings to human aging.

## Figures and Tables

**Figure 1 biomolecules-14-01079-f001:**
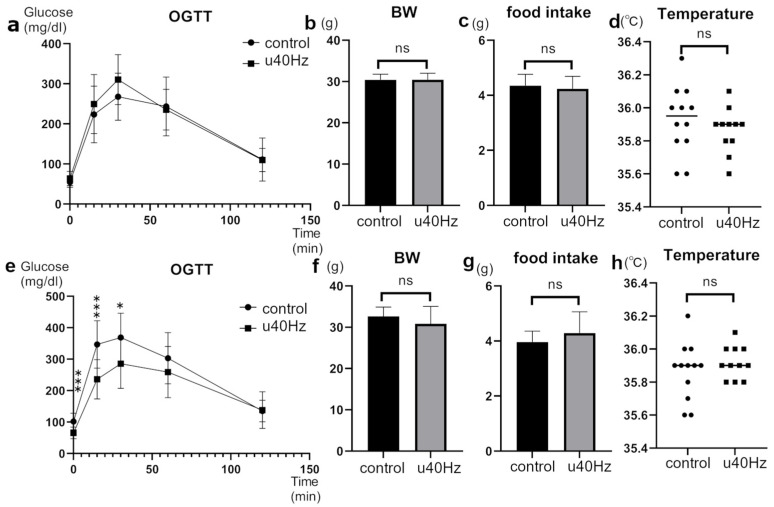
Effect of u40Hz stimulation on glucose tolerance in SAMP-10 mice. Measurements were taken at 20 weeks of age (pre-40Hz stimulation) and at 31 weeks of age (post-40Hz stimulation). The u40Hz-stimulated group (*n* = 11) and control group (*n* = 12) were compared. Panels (**a**–**d**) show pre-stimulation data: (**a**) oral glucose-tolerance test results, (**b**) food intake, (**c**) body weight, and (**d**) body temperature. Panels (**e**–**h**) show post-stimulation data: (**e**) OGTT results, (**f**) food intake, (**g**) body weight, and (**h**) body temperature. *** *p* < 0.001, * *p* < 0.05, ns indicates no significant difference.

**Figure 2 biomolecules-14-01079-f002:**
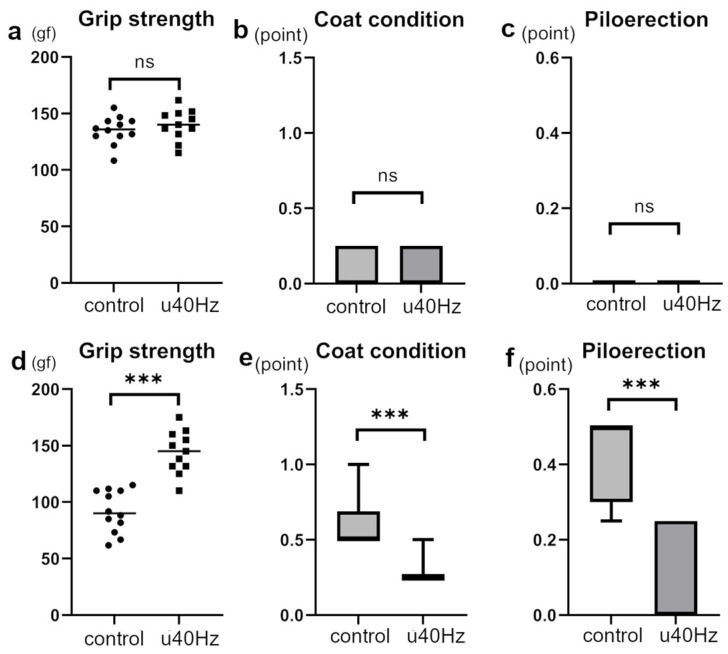
Assessment of specific frailty parameters in SAMP-10 mice pre- and post-u40Hz stimulation. The evaluations were based on a 31-item frailty index (FI) scale, highlighting significant changes in specific items. Measurements were taken at 20 weeks of age (pre-40Hz stimulation) and at 31 weeks of age (post-40Hz stimulation). The u40Hz-stimulated group (*n* = 11) and control group (*n* = 12) were compared. Panels (**a**–**c**) show pre-stimulation data at 20 weeks: (**a**) grip strength, (**b**) coat condition, and (**c**) piloerection. Panels (**d**–**f**) show post-stimulation data at 31 weeks: (**d**) grip strength, (**e**) coat condition, and (**f**) piloerection. *** *p* < 0.001, ns indicates no significant difference.

**Figure 3 biomolecules-14-01079-f003:**
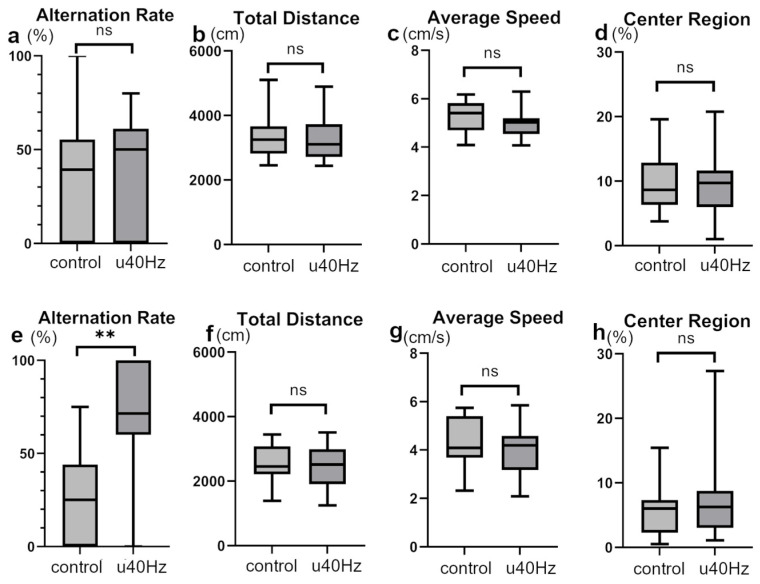
Assessment of cognitive and behavioral parameters in SAMP-10 mice pre- and post-u40Hz stimulation. Measurements were taken at 20 weeks of age (pre-40Hz stimulation) and at 31 weeks of age (post-40Hz stimulation). The u40Hz-stimulated group (*n* = 11) and control group (*n* = 12) were compared. Panels (**a**–**d**) show pre-stimulation data at 20 weeks: (**a**) Y-maze test alternation rate, (**b**) total distance traveled in the open-field test, (**c**) average speed in the open-field test, and (**d**) time spent in the center region in the open-field test. Panels (**e**–**h**) show post-stimulation data at 31 weeks: (**e**) Y-maze test alternation rate, (**f**) total distance traveled in the open-field test, (**g**) average speed in the open-field test, and (**h**) time spent in the center region in the open-field test. ** *p* < 0.01, ns indicates no significant difference.

**Table 1 biomolecules-14-01079-t001:** Comparison of frailty index (FI) between control and u40Hz-stimulated SAMP-10 mice after 5 weeks of stimulation.

	Control (*n* = 12)		u40Hz (*n* = 11)		*p* Value
	Median	IQR	Median	IQR	
Pre-u40Hz stimulation	0.25	0.69	0.25	0.3	0.83
Post-u40Hz stimulation	3.23	1.43	2.25	1	0.02 *

FI assessments were conducted at 25 weeks of age (pre-stimulation) and at 31 weeks of age (post-stimulation) after 5 weeks of u40Hz stimulation. The table shows the median FI scores for control mice (*n* = 12) and u40Hz-stimulated mice (*n* = 11). * *p* < 0.05.

## Data Availability

Microarray data were deposited in the National Center for Biotechnology Information Gene Expression Omnibus (https://www.ncbi.nlm.nih.gov/geo/ (accessed on 27 August 2024)). accession number: GSE269209.

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
