# Peer review of "Unmodulated 40 Hz Stimulation as a Therapeutic Strategy for Aging: Improvements in Metabolism, Frailty, and Cognitive Function in Senescence-Accelerated Prone 10 Mice"

_biomolecules, 2024, doi:10.3390/biom14091079_

Round 1

Reviewer 1 Report

Comments and Suggestions for Authors

Concerns to the authors.

1.    Materials and Methods: -How mice were assigned to the study?

2.    Most of the studies used 30 min-1h of applied vibration per day. This study used 24 h vibration technique. How mice were behaving after this? Is it feasible? It would be useful if authors would give more explanation on this part.

3.    How food intake experiment was performed? Authors need to add the description to Materials and Methods.

4.    Results: I do not follow results/figures for all “before stimulation tests”. All parameters should be not different in any performed test. If authors expected difference in the paramers they should explain why.

5.  Fig1. OGTT- authors should indicate time points when they have measured glucose. Regarding statistical analysis-Two Way Anova is more proper method than t-test.

6.    Line 206-Table 1 is missing.

7.    Results in Supplementary material – FigS1, “Changes in Gene Expression in the PPARγ Signaling Pathway” is difficult to visualize and follow. Perhaps authors will change how they show these results and present them in the table.

8.    Discussion: The novelty and limitations of the findings needs to be stated and discussed.

Author Response

Response to Reviewer 1 Comments

Point-by-point response to Comments and Suggestions for Authors

Comments 1:

Materials and Methods: -How mice were assigned to the study?

Response 1:

Thank you for pointing this out. The SAMP-10 mice were chosen for this study because they are an accelerated aging model, not due to a specific genetic mutation but rather due to multiple genetic abnormalities. This makes them more representative of natural aging processes. Additionally, SAMP-10 mice are frequently used as a model for cognitive decline, which aligns with our study's focus on investigating muscle changes associated with cognitive impairment. Therefore, we considered them well-suited for our experimental setup.

Comments 2:

Most of the studies used 30 min-1h of applied vibration per day. This study used 24 h vibration technique. How mice were behaving after this? Is it feasible? It would be useful if authors would give more explanation on this part.

Response 2:

Thank you for your insightful comments regarding the duration of vibration exposure used in our study. As noted, many previous studies have applied vibration for 30 minutes to 1 hour per day. In our study, we used a 24-hour vibration technique, and we found no adverse effects on the mice, indicating that it is feasible. We believe that short durations of 30 minutes to 1 hour may not induce the significant brain changes reported in other studies. Furthermore, in our previous research, we confirmed that 24-hour stimulation for three months in humans also did not result in any adverse effects.

( https://www.neuroscigroup.us/articles/AADC-5-116.php)

Comments 3:

 How food intake experiment was performed? Authors need to add the description to Materials and Methods.

Response 3:

Thank you for pointing this out. We will add more details to the Materials and Methods section regarding the methodology of the food intake experiment as follows.(Line 125-133)

To assess food intake, we measured the reduction in the amount of food over a three-day period. The experimental procedure was as follows: Initially, we measured and recorded the weight of the food provided to each mouse at the beginning of the experiment. The mice were then allowed to consume food freely for a period of three days. At the end of this period, we weighed the remaining food. The total amount of food consumed over the three days was calculated by subtracting the final weight from the initial weight. This total was then divided by three to obtain the average daily food intake for each mouse.

Comments 4:

Results: I do not follow results/figures for all “before stimulation tests”. All parameters should be not different in any performed test. If authors expected difference in the paramers they should explain why.

Response 4:

Thank you for your valuable insights. The behavioral science experiments conducted in this study, such as glucose tolerance tests including blood glucose levels, flail evaluation, and cognitive function evaluation, are experimental systems that are very prone to individual differences. Therefore, we believe that it is very important to show these results in order to guarantee that the two groups we have divided this time are appropriately grouped without significant differences before stimulation.

Comments 5:

Fig1. OGTT- authors should indicate time points when they have measured glucose. Regarding statistical analysis-Two Way Anova is more proper method than t-test.

Response 5:

Thank you for your valuable feedback on our manuscript. We appreciate the suggestions and have made the following revisions:

1. Time Points for OGTT Measurements:

We have added the specific time points at which glucose measurements were taken during the oral glucose tolerance test (OGTT) to the Materials and Methods section (Line119-124). These time points are fasting (0 min), 15 min, 30 min, 60 min and 120 min post-load.

2. Statistical Analysis:

We have revised our analysis using Two Way ANOVA as suggested. Our findings are as follows:

Before the u40Hz stimulation, the two-way ANOVA indicated only a significant inter-action effect with time between the two groups (P < 0.001). After the u40Hz stimulation, we observed a significant main effect due to the u40Hz stimulation (P = 0.028) and a significant interaction effect with time (P < 0.001). The combined interaction effect of u40Hz stimulation and time approached significance (P = 0.053).

 We have added the above to the Results section (Line200-204).

We have added information about the two-way ANOVA in the Statistical Analyses section of the Materials and Methods (Line185-186).

Comments 6:

Line 206-Table 1 is missing.

Response 6:

Thank you for pointing out the missing Table 1. I apologize for the oversight. It was an error on my part, and I neglected to include Table 1 in the manuscript. I will include Table 1 in the appropriate section of the manuscript and resubmit it for your review. Thank you for your understanding and patience.

Comments 7:

Results in Supplementary material – FigS1, “Changes in Gene Expression in the PPARγ Signaling Pathway” is difficult to visualize and follow. Perhaps authors will change how they show these results and present them in the table.

Response 7:

Thank you for your valuable feedback regarding Figure S1. In response to your comment, we have made the following changes to improve the visualization and clarity of the data:

1.Enhanced Visualization of Key Differences:

We have retained the signaling pathway but have zoomed in on the areas where significant differences were observed between the control group and the u40Hz stimulation group. This allows for a clearer and more focused comparison of the relevant changes in gene expression.

2.Cluster Analysis Heatmap:

To provide an overall view of the changes and trends, we have created a heatmap resulting from the cluster analysis of the microarray data. This heatmap offers a comprehensive visual summary of the gene expression changes, facilitating a better understanding of the overall data patterns.

3.Clearer Presentation Format:

In addition to the visual enhancements, we have organized the data into a table format where appropriate. This allows for a more straightforward comparison and interpretation of the results.

Comments 8:

Discussion: The novelty and limitations of the findings needs to be stated and discussed.

Response 8:

Thank you for your valuable feedback. We appreciate the opportunity to clarify the novelty and limitations of our study.

The novelty of our findings lies in the application of u40Hz stimulation beyond its well-documented effects on cognitive function improvement and amyloid-beta clearance through aquaporin-4, as recently reported (Nature. 2024 Mar;627(8002):149-156.). Our study is the first to demonstrate the positive impact of u40Hz stimulation on glucose tolerance and muscle preservation, which are significant aspects of aging that have not been previously explored. However, there are several limitations to our study. We used SAMP-10 mice, which are a model of accelerated aging. It will be essential to verify these findings in naturally aging mice, frail mice, and diabetic model mice in future research. Additionally, while u40Hz stimulation has shown benefits in other studies, the effects of other lower frequencies should be investigated. Furthermore, the results of our microarray analysis need to be validated through detailed cellular-level studies. We will include these points in the Discussion section of the paper (Line360-370).

Reviewer 2 Report

Comments and Suggestions for Authors
  • A brief summary

The authors utilized behavioral and microarray methods to study the beneficial effects of the Unmodulated 40-Hz stimulation. The most interesting data from microarray is presented in an unaccepted way.

  • General concept comments
    Microarray was done on the triceps brachii muscle from the mice. However, no microarray data was shown in the main manuscript. Microarray is a high-throughput method that can give way more results than the authors presented in the supplemental materials. Another concern about the Microarray is the tissue source. Regarding cognitive improvements, so why was only muscle included?
  • Specific comments 

1. The dimensions of the open field and Y-maze, and the video record and analysis software should be listed in the method session.

2. Line 171: Results of the glucose tolerance test are a two-factor dataset; two-way ANOVA is a more meaningful test.

3. The glucose tolerance results for pre- and post-stimulation in Figure 1A and Figure 1E: It seems the differences come from the control group rather than the stimulation. The pre-stimulation control was ~220, and the post-stimulation control was 350 at 15 min post-load. How do we explain the dramatic increase for the control group six weeks later?

4. Line 206 and 207: no table is available in the manuscript.

5. Figure 2a and Figure 2d show similar concerns about the control group as in glucose tolerance results. The difference comes from the control group; the only factor is that the post-stimulation control group is 6 weeks older than the pre-stimulation control. The 6-week duration is critical in this comparison; how do you explain it?  Any data to support this finding?

6.  Figure 3: the y-maze usually shows a spontaneous alternation above 50% for normal mice. However, the number is only 25% for the control group in this study.

Author Response

Response to Reviewer 2 Comments

Point-by-point response to Comments and Suggestions for Authors

A brief summary:

The authors utilized behavioral and microarray methods to study the beneficial effects of the Unmodulated 40-Hz stimulation. The most interesting data from microarray is presented in an unaccepted way.

Response:

Thank you for your insightful comments regarding the presentation of our microarray data. We appreciate the feedback and have taken steps to improve the clarity and acceptability of our results presentation. In response to your concerns, we have made the following modifications:

1.Enhanced Visualization of Key Differences:

We have focused on the significant differences in gene expression between the control group and the u40Hz stimulation group by zooming in on these areas within the signaling pathway. This approach provides a more detailed and focused view of the most relevant changes.

2.Cluster Analysis Heatmap:

To give a comprehensive overview of the changes, we have created a heatmap from the cluster analysis of the microarray data. This heatmap illustrates the overall patterns and trends in gene expression changes, making the data easier to interpret and follow.

3.Clearer Presentation Format:

In addition to the visual enhancements, we have organized the data into a table format where appropriate. This allows for a more straightforward comparison and interpretation of the results.

General concept comments:

Microarray was done on the triceps brachii muscle from the mice. However, no microarray data was shown in the main manuscript. Microarray is a high-throughput method that can give way more results than the authors presented in the supplemental materials. Another concern about the Microarray is the tissue source. Regarding cognitive improvements, so why was only muscle included?

Response:

Thank you for your insightful comments. We conducted the microarray analysis on the triceps brachii muscle because our glucose tolerance test and muscle strength measurements indicated significant differences in this tissue. Therefore, we prioritized examining the muscle for initial changes. However, we recognize the importance of understanding the mechanisms underlying cognitive improvements. We are currently analyzing the mouse brain to investigate the factors contributing to these improvements, and we will include these findings in future studies.

Comments 1:

The dimensions of the open field and Y-maze, and the video record and analysis software should be listed in the method session.

Response 1:

Thank you for your insightful comments. We have provided additional details in the Materials and Methods section (Line149, Line153, Line163-166) as follows:

Dimensions of the Open Field and Y-maze:

Open field: 500W x 500Dx 300Hmm

Y-maze: corridor floor width=30W, corridor upper part width=120W x 400D x 120Hmm

Video Recording and Analysis Software:

Software for Open field: TimeOFCR1 (O' HARA & CO., LTD.)

Software for Y-maze: TimeYM1 (O' HARA & CO., LTD.)

Comments 2:

 Line 171: Results of the glucose tolerance test are a two-factor dataset; two-way ANOVA is a more meaningful test.

Response 2:

Thank you for your valuable feedback on our manuscript. We agree with your recommendation regarding the statistical analysis. We have re-analyzed the data using a Two-Way ANOVA,

Our findings are as follows:

Before the u40Hz stimulation, the two-way ANOVA indicated only a significant inter-action effect with time between the two groups (P < 0.001). After the u40Hz stimulation, we observed a significant main effect due to the u40Hz stimulation (P = 0.028) and a significant interaction effect with time (P < 0.001). The combined interaction effect of u40Hz stimulation and time approached significance (P = 0.053).

We have added the above to the Results section (Line200-204).

We have added information about the two-way ANOVA in the Statistical Analyses section of the Materials and Methods (Line185-186).

Comments 3:

The glucose tolerance results for pre- and post-stimulation in Figure 1A and Figure 1E: It seems the differences come from the control group rather than the stimulation. The pre-stimulation control was ~220, and the post-stimulation control was 350 at 15 min post-load. How do we explain the dramatic increase for the control group six weeks later?

Response 3:

Thank you for your insightful question regarding the glucose tolerance results in Figure 1A and Figure 1E.

The glucose tolerance tests were conducted at different time points to assess the effects of u40Hz stimulation over time. Specifically, the pre-stimulation values were obtained five weeks before the start of the u40Hz stimulation, while the post-stimulation values were measured after five weeks of continuous stimulation, making the post-stimulation data reflect changes over a total period of 11 weeks.

The SAMP-10 mice used in this study are a model of accelerated aging, characterized by a rapid decline in various physiological functions, including glucose metabolism. The observed deterioration in glucose tolerance in the control group, evidenced by the increase from ~220 mg/dL to 350 mg/dL at 15 minutes post-load over the 11-week period, is within the expected range for this mouse model. This degree of decline is not beyond what is anticipated for SAMP-10 mice over this duration.

The accelerated aging model is known to exhibit significant metabolic changes over relatively short periods, and the observed decline in glucose tolerance in the control group is within the expected range of aging-related metabolic decline. The u40Hz stimulation, on the other hand, appears to mitigate this decline, as evidenced by the significantly lower blood glucose levels in the stimulation group compared to the control group at various time points post-load.

Comments 4:

Line 206 and 207: no table is available in the manuscript.

Response 4:

Thank you for pointing out the missing Table 1. I apologize for the oversight. It was an error on my part, and I neglected to include Table 1 in the manuscript. I will include Table 1 in the appropriate section of the manuscript and resubmit it for your review. Thank you for your understanding and patience.

Comments 5:

Figure 2a and Figure 2d show similar concerns about the control group as in glucose tolerance results. The difference comes from the control group; the only factor is that the post-stimulation control group is 6 weeks older than the pre-stimulation control. The 6-week duration is critical in this comparison; how do you explain it?  Any data to support this finding?

Response 5:

Thank you for your valuable insights. Although the decrease in muscle strength in the control group appears somewhat pronounced, we believe it reflects the age-related changes in accelerated aging mice. Previous study has shown that grip strength in 38-week-old SAMP-10 mice decreases to about one-third of the control group (Sci Rep. 2022 Feb 14;12(1):2425). We interpret the results as indicating that while muscle strength typically declines with age, the u40Hz stimulation has helped mitigate this decline.

Comments 6:

Figure 3: the y-maze usually shows a spontaneous alternation above 50% for normal mice. However, the number is only 25% for the control group in this study.

Response 6:

Thank you for your valuable feedback. The SAMP-10 mice used in this study are an accelerated aging model, which leads to an earlier onset of cognitive decline compared to normal mice. This earlier decline is responsible for the reduced alternation rate observed.
